# An Evaluation of Antibiotic Prescribing Practices in a Rural Refugee Settlement District in Uganda

**DOI:** 10.3390/antibiotics10020172

**Published:** 2021-02-09

**Authors:** Matua Bonniface, Winnie Nambatya, Kalidi Rajab

**Affiliations:** 1Department of Health, Yumbe District Local Government, 60811 Yumbe, Uganda; bonifacematua@yahoo.com; 2East African Community Regional Center of Excellence for Vaccines, Immunizations and Health Supply Chain, Kicukiro Campus, School of Public Health, University of Rwanda, Kigali, Rwanda; 3Department of Pharmacy, Makerere University, University Rd, 10218 Kampala, Uganda; wnambatya@chs.mak.ac.ug

**Keywords:** antibiotics, prescribing practices, availability

## Abstract

Ensuring access to effective antibiotics and rational prescribing of antibiotics are critical in reducing antibiotic resistance. In this study, we evaluated antibiotic prescribing practices in a rural district in Uganda. It was a cross-sectional study that involved a retrospective review of 500 outpatient prescriptions from five health facilities. The prescriptions were systematically sampled. World Health Organization core medicine use prescribing and facility indicators were used. Percentage of encounters with one or more antibiotics prescribed was 23% (10,402/45,160). The mean number of antibiotics per prescription was 1.3 (669/500). About 27% (133/500) of the diagnoses and 42% (155/367) of the prescriptions were noncompliant with the national treatment guidelines. Prescribing antibiotics for nonbacterial infections such as malaria 32% (50/156) and noninfectious conditions such as dysmenorrhea and lumbago 15% (23/156) and nonspecific diagnosis such as respiratory tract infection 40% (59/133) were considered noncompliant with the guidelines. On average, 68% (51/75) of the antibiotics were available on the day of the visit. Inappropriate prescribing practices included excessive use of antibiotics and failure to diagnose and prescribe in compliance with treatment guidelines. There is a need to strengthen antibiotic use in the health facilities through setting up stewardship programs and interventions to promote adherence to national treatment guidelines.

## 1. Introduction

Antibiotic resistance (ABR) has recently been rising steadily worldwide and has reduced the ability of antibiotics to effectively control infectious diseases [1]. ABR causes 700,000 deaths annually across the globe, a number that is projected to increase to 10 million by the year 2050 if new interventions are not developed [2]. Data published from Uganda show an increase in the trends of ABR [3]. The prevalence of Methicillin-Resistant *Staphylococcus Aureus* (MRSA) varied from 2–50%, while Extended Spectrum Beta-lactamase (ESBL) prevalence ranged from 10–75% [4]. In addition, increasing resistance, ranging from 4–30%, has been reported among Gram-negative enterobacteria against carbapenems, a last-line treatment, and a broad range of bacteria have still shown high levels of resistance (over 50%) in many cases to commonly used antibiotics such as penicillin, tetracyclines, and co-trimoxazole [4]. In line with this, the United Nations (UN) and the World Health Organization (WHO) have called for actions to this public health threat [5].

The main driver of ABR is misuse of antibiotics, aggravated by other factors such as unrestricted access to antibiotics [6]. Common examples of irrational antibiotic use include incorrect diagnosis; prescribing of antibiotics for nonspecific conditions such as childhood diarrhea, mild, nonbacterial infection, upper respiratory tract infection, and simple malaria; polypharmacy; over-prescribing; excessive or unnecessary use of injections; improper use of antibiotics by patients; failure to prescribe in compliance with treatment guidelines; and the use of antibiotics as livestock food additives for growth promotion [7,8,9,10,11]. The consequences of such irrational antibiotic use include poor or limited quality of care, high cost of therapy, low availability, and increased incidence of adverse effects such as prolonged morbidity, mortality, drug toxicity, prolonged hospitalization, microbial antibiotic resistance, and the associated resistant infections [12,13,14,15].

Studies in Uganda show that most of the medicine use problems are caused by irrational prescribing practices [16]. However, most of these studies assessed prescribing practices generally or prescribing for specific health conditions such as upper respiratory tract infections, malaria, and simple diarrhea [17,18,19,20,21]. Little is known about prescribing practices for antibiotics in Yumbe, a refugee settlement and border district, and Uganda at large. We evaluated the prescribing practices of antibiotics in all the level 3 and 4 health facilities in Yumbe with the aim of providing information useful in designing interventions to improve appropriate use of antibiotics so as to achieve optimal clinical outcomes related to antibiotic use, minimizing toxicity and other adverse events, reducing the costs of health care for infections, and limiting the selection for antibiotic-resistant strains.

## 2. Results

In the review period, a total of 10,402 (23%) prescriptions of the 45,160 prescriptions contained an antibiotic. The 500 sampled prescriptions of the 10,402 prescriptions with at least an antibiotic prescribed contained 669 antibiotics.

### 2.1. Prescribing Practices

The common diagnoses for which antibiotics were prescribed were respiratory tract infections (117, 23%), urinary tract infections (84, 17%), and malaria (50, 10%) (Table 1).

The most commonly prescribed antibiotics were penicillins (324, 48%), nitroimidazoles (108, 16%), quinolones (58, 9%), cephalosporins (56, 8%), and aminoglycosides (47, 7%) (Table 2).

Percentage of encounters with one or more antibiotics prescribed was 23% (10,402/45,160) with 55% (369/669) of the antibiotics prescribed by generic name. On average, 1.3 (669/500) antibiotics were prescribed per patient and 29% (144/500) of the patients received two or more antibiotics. Of the total number of patients, 125 (25%) received antibiotic injections. All the medicines prescribed were from the Essential Medicines and Health Supplies List (EMHSL) of Uganda. The average cost of antibiotics prescribed per patient was USD 0.5. Antibiotics were prescribed for an average duration of 4.8 days (3–10), with majority of patients 440 (88%) receiving antibiotics for a duration of five days. Majority of the prescriptions, 485 (97%), had diagnosis of the patient recorded. However, 133 (27%) of the diagnoses did not comply with the Uganda Clinical Guidelines (UCG) and were, therefore, considered inappropriate. Of the 367 (73%) of the diagnoses that were compliant with the guidelines, 155 (42%) of the antibiotics prescribed for the diagnosis were not compliant with UCG. (Table 3, Appendix A).

The main reasons for noncompliance of the prescriptions with the guidelines included prescribing antibiotics for malaria (50, 32%); helminthiasis (worm infestation) (17, 11%); dental conditions (19, 12%) such as pulpitis and gingivitis, for which antibiotics are not indicated; and noninfectious conditions such as dysmenorrhea and lumbago (23, 15%) (Table 4). The diagnoses that were considered noncompliant with the guidelines included nonspecific and vaguely written diagnosis such as respiratory tract infections (59, 44%), upper respiratory tract infections (40, 30%), gastroenteritis (15, 11%), and bacterial infection (10, 8%) (Table 5).

### 2.2. Health Facility Factors Affecting Prescribing Practices

All the facilities had UCG and no EMHSL. Only two (40%) facilities had diagnostic laboratory facilities for complete blood count, and these were the health center grade IVs (HCIVs). On average, 68% (50/75) of the antibiotics were available on the day of the visit. No health facility had all the selected antibiotics available on the day of the visit and two injectable antibiotics, penicillin benzyl and penicillin procaine fortified (PPF), were available throughout the review period. The average number of days the antibiotics were out of stock was eight days. Cloxacillin was most frequently out of stock with only one facility that did not register a stock out in the review period. The percentage cost contribution of antibiotics was 38% (11,904,870 Ugandan shillings, about 3200 USD) of total expenditure on essential medicines (31,673,655 Ugandan shillings, about 8500 USD). The percentage contribution for higher-level facilities (35% and 27%) was less than that of lower-level facilities (44%, 51%, and 56%). (Table 6, Appendix A).

## 3. Discussion

In order to design effective interventions to improve appropriate use of antibiotics, it is important to understand prescribing practices of antibiotics. This study provides an understanding of these prescribing practices.

The findings of this survey revealed high percentage of patients receiving antibiotics and low percentage of prescribing by generic name. Studies done on antibiotic prescription patterns in a Ghanaian primary health care facility and Khartoum state found a higher percentage encounter of one or more antibiotics prescribed, of 29% and 54%, respectively [22,23]. This difference could also be attributed to a limited range of antibiotics available in public health facilities in Uganda. However, our findings are within WHO standard 20–26% and more than the national standard ≤15% [24,25,26]. This high exposure to antibiotics means high chances of development of resistance. The prescription by generic name is well below the WHO standard of 100% and national standard of ≥85% [24,25,26]. Previous studies conducted in Cameroon and Khartoum state found 98% and 37% antibiotics were prescribed by generic names, respectively [22,23]. Prescribing by generic name reduces dispensing errors, promotes patient understanding of medicines, and prevents extravagant prescribing, hence improving medicine use and should, therefore, be encouraged among the prescribers. The average number of antibiotics per prescription was lower than that reported in a study carried out at a tertiary teaching care hospital, Gujrat, of 1.5 and WHO standard of 1.6–1.8 [25,26,27]. It is preferable to keep the mean number of antibiotics per prescription as low as possible so as to prevent risk of drug–drug interaction and reduce cost of treatment, pill burden, and out of stock of antibiotics. This will in turn reduce exposure to antibiotics and promote adherence, slowing the development of antibiotic resistance.

The average cost of antibiotics prescribed per patient ($0.5) was higher than per capita allocation for medicines ($0.3). The expenditure on antibiotic medicines as a percentage of total facility medicine costs was also high (38%). These show the health facilities operated with small budgets for medicines and spent more on antibiotics. The lower health facilities that receive kits (push system) had higher expenditure on antibiotics compared with higher levels that determine their own needs (pull system). Adding to this expenditure was the high percentage of patients receiving parenteral antibiotics more than the WHO standard for percentage of encounters, with an injection prescribed 13–24% and national standard of less than or equal to 15% [24,25,26]. More so, expensive parenteral antibiotics such as ceftriaxone were also commonly used. Unnecessary use of parenteral antibiotics adds to cost of therapy and also increases the risk of blood-borne infections and other complications associated with use of injections. Therefore, improving antibiotic use, thereby reducing antibiotic consumption, can reduce costs on antibiotics and increase antibiotic availability. The availability of antibiotics was low. No health facility had all the selected antibiotics available. Some of the patients did not have their prescriptions filled if antibiotics were unavailable, hence failure to get treatment. This can lead to resistance, death, or other complications and out-of-pocket expenditure, which the individuals may not afford, being a poor community. Non-availability of antibiotics also leads to loss of confidence by the patients in the health care facility and system and can affect prescribing patterns, as the prescribers may be forced to prescribe what is available but not what is appropriate for the clinical condition.

Adherence to UCG was found to be suboptimal. Recording of diagnosis before prescriptions was above the national target of 85%, though lower than WHO standard of 100% [24,25,26]. Respiratory tract infections and urinary tract infections were the main diagnoses for which antibiotics were prescribed. These correlate with the prescribing patterns of the antibiotics. For example, the UCG recommends amoxicillin for non-severe pneumonia and ceftriaxone for severe pneumonia, while ciprofloxacin is recommended for urinary tract infections including acute cystitis and prostatitis. Only 73% of the antibiotic prescriptions had the diagnosis properly written following UCG despite all the facilities having UCG. The diagnoses that were considered noncompliant with the guidelines included vaguely written and nonspecific diagnoses such as respiratory tract infections/upper respiratory tract infections, bacterial infections, and gastroenteritis. This can affect use of the treatment guidelines, reporting of prevalence/incidence of diseases, and antibiotic use. For example, it is difficult to tell whether respiratory tract infection is viral or bacterial and, therefore, warrants antibiotic or not. More so, each diagnosis in the guidelines has an International Classification of Diseases 10th version (ICD10) code and documentation and reporting are based on this code. The noncompliance may be as a result of knowledge gaps and diagnostic uncertainty and it may also imply that the treatment guidelines are not being referred to while prescribing medicines. In fact, 42% of the prescriptions did not comply with UCG and only the two higher-level health facilities had diagnostic laboratory equipment for complete blood count. The main reasons for noncompliance of the prescriptions with the guidelines included prescribing antibiotics for malaria, helminthiasis (worm infestation), dental conditions such as pulpitis and gingivitis for which antibiotics are not indicated, and noninfectious conditions such as dysmenorrhea. Following the UCG 2016, Ministry of Health Uganda, no antibiotic medicines should be administered as treatment for those conditions [28]. This demonstrates antibiotic misuse, which can potentially lead to increased antibiotic resistance, thus increasing the necessity to use more expensive antibiotics to treat life-threatening infections caused by resistant bacteria in the future. As observed by Ofori-Asenso and Agyeman, overuse of antibiotics for nonspecific conditions such as childhood diarrhea, mild nonbacterial infection, upper respiratory tract infection, and simple malaria contributes to antibiotic resistance [11]. These irrational prescribing patterns may be a result of inadequate knowledge and skills in prescribing antibiotics, lack of diagnostic facilities, and low cadre prescribers. A study conducted in Uganda indicated that the majority of prescribing cadres were lower-cadre professionals such as nursing assistants and enrolled nurses [29]. The lower health facilities without diagnostic laboratory equipment may not be able to appropriately diagnose infections since most of their diagnoses will be based on clinical signs and symptoms. This further perpetuates prescribing to be on the safe side.

As a limitation, these indicators highlight major problem areas of antibiotic use patterns and quantify the magnitude of the problem at glance, but they do not exhaustively answer why the problem existed. The study was also limited only to outpatients and, therefore, could not give overall antibiotic prescribing patterns in the facilities.

## 4. Materials and Methods

This study was conducted in Yumbe district located in the northwestern part of Uganda, located approximately 590 km from Kampala, Uganda’s capital city. The district total population is 485,582, of which 48% are males and 52% are females. The district hosts refugees mainly from South Sudan and the Democratic Republic of Congo. There are 27 health facilities in the district. These include a general hospital (Yumbe hospital), which is the district referral hospital, two HC IVs, seven (7) health center grade IIIs (HC IIIs), and 17 health center grade IIs (HC IIs). These health facilities provide both curative and preventive services including Outpatient, in patient, Maternal and child health services, and community outreaches, plus other specialized services depending on the level of the facility. The study was conducted in five Health facilities: Yumbe Hospital/Yumbe HC IV, Midigo HC IV, Ariwa, Barakala, and Kulikulinga HC IIIs. These Health facilities were purposely selected. They attend to refugees and have admission facilities and laboratories. Yumbe Hospital was under major renovation and its services were transferred to Yumbe HC IV during the data collection period.

This was a cross-sectional study. Outpatient records for a period of three months from March–May 2019 were reviewed. WHO core medicine use indicators for assessing outpatient medicine use were used.

The study population included outpatients who were prescribed antibiotics from the five health facilities and tracer antibiotics.

Prescriptions containing systemic antibiotics for both adults and children were considered for inclusion, while prescriptions containing topical antibiotics such as lotions, ointments, vaginal pessaries, and eye preparations like eye drops were excluded. Also, prescriptions from outreach and special doctors’ clinics were excluded. Only the selected 15 tracer antibiotics were used to assess for availability. Antibiotics in this study refer to medicines for treatment of bacterial infections only.

Based on WHO/DAP/93 recommendation for sample size determination and sampling technique for prescribing indicators, 100 prescriptions were taken from each corresponding Health facility [30], meaning a total of 500 prescription forms meeting the inclusion criteria were reviewed in this study. A systematic sampling technique was employed to select the 100 prescriptions from each Health facility. The total numbers of prescriptions in the review period with an antibiotic prescribed were determined and the sampling interval was determined by dividing the total number by 100. A simple random sampling was used to select the first prescription. For Health facility indicators, 15 key tracer antibiotics were selected from each facility as per WHO recommendation of a minimum of 15 essential tracer medicines in each health facility. These were purposely selected by only including antibiotics that are expected to be available at the lowest level of care of the study facilities, i.e., HC III, according to the national EMHSL.

The study indicators included:Prescribing indicators;Percentage of encounters with one or more antibiotics prescribed;Percentage of antibiotics prescribed by generic name;Average number of antibiotics prescribed per patient;Percentage of patients prescribed antibiotic medicines with diagnosis recorded;Percentage of patients receiving antibiotic injections;Percentage of antibiotics prescribed, consistent with the EMHSL;Average cost of antibiotics prescribed per patient;Average duration of prescribed antibiotic treatment;Percentage of prescriptions in accordance with the Standard Treatment Guidelines (STGs);Facility indicators/factors;Existence of STGs for infectious diseases;Existence of an approved hospital formulary list or essential medicines’ list (EML);Availability of a set of key antibiotics in the facility stores on the day of the study;Average number of days that a set of key antibiotics is out of stock;Expenditure on antibiotics as a percentage of total facility medicine costs; andAvailability of diagnostic laboratory facility (complete blood count).

Data were collected using a structured check list for prescribing and health facility indicators. Data regarding prescribing indicators were taken from sampled prescription records retrospectively and were filled or recorded in the structured check list accordingly. Additionally, the availability of tracer antibiotics, which were assessed from the store, and the presence of EML and STG in the Outpatient Department (OPD) were also assessed in the facility indicator form accordingly. Data were collected from OPD registers, Dispensing logs, and stock cards or stock book. Additional information (e.g., prices) was obtained from recent invoices. Data collection was supervised by the researchers.

Microsoft Excel 2010 version was used for the analysis. The scores of the indicators were compared with national and WHO standards/targets.

Makerere University School of Health Sciences Research Ethics committee (MU-SHS-REC) granted ethical approval for the study.

## 5. Conclusions

Irrational prescribing practices observed in this study included a high number of encounters with an antibiotic prescribed, low percentage of prescribing by generic name, higher percentage of patients receiving injectable antibiotics, and low compliance with standard treatment guidelines even though the treatment guidelines where available in all the facilities. Most of the diagnoses did not comply with the treatment guidelines and antibiotics were prescribed for indications such as uncomplicated malaria, helminthiasis, and allergy that do not require antibiotic interventions in their standard management guides. The facility factors that could have affected the prescribing practices included low availability, out-of-stock medicines, and lack of diagnostic laboratory facilities. These can lead to suboptimal therapeutic outcomes and perpetuate development of drug resistance.

There is need to strengthen antibiotic use in health facilities and hospitals through setting up stewardship programs and interventions to enforce the national standard treatment guidelines and provision of diagnostic facilities. More emphasis needs to be placed on teaching the art of writing a prescription to prescribers and compliance with standard treatment guidelines. There is also need to improve availability of the antibiotics. We recommend future studies on the outcomes of antibiotic treatment, underlying causes of irrational antibiotic use, and antibiotic resistance patterns in these facilities.

## Figures and Tables

**Table 1 antibiotics-10-00172-t001:** Showing diagnosis for which antibiotics were prescribed.

Diagnosis	Frequency (*n* = 500)	Percentage (%)
Respiratory tract infection	117	23
Urinary tract infection	84	17
Malaria	50	10
Gastroenteritis	42	8
Pelvic inflammatory disease	29	6
Peptic ulcer disease	26	5
Bacterial infection/septicemia	19	4
Helminthiasis	16	3
Otitis media	11	2
Septic wound/abscess	9	2
Gingivitis	8	2
Allergy/insect bite	8	2
Dog bite	8	2
Others	73	15

Others: Skin rash, diarrhea, dysmenorrhea, lumbago, trauma, candidiasis, wound, appendicitis, somatic pain, hernia, asthma, burn, oral sores, breast enlargement, pulpitis, measles, dermatitis, severe fibroids, and fatigue.

**Table 2 antibiotics-10-00172-t002:** Showing class of antibiotics prescribed.

Class of Medicine	Frequency (*n* = 669)	Percentage
Penicillins	324	48
Nitroimidazoles	108	16
Fluoroquinolones	58	9
Cephalosporins	56	8
Aminoglycoside	47	7
Sulphonamides	26	4
Tetracyclines	29	4
Macrolides	16	2
Nitrofurantoin	5	1

Penicillins (Amoxycillin 24%, Penicillin Procaine Fortified 11%, Ampicillin+amoxicillininj 7%, Penicillin benzyl 6%), Nitroimidazoles (metronidazole 16%), Fluroquinolones (Ciprofloxaxin 9%), cephalosporins (Ceftriaxone 8%), aminoglosides (Gentamycin 7%), sulfonamides (Cotrimoxazole 4%), Macrolides (Erythromycin 2%).

**Table 3 antibiotics-10-00172-t003:** Showing results for prescribing indicators.

Prescribing Indicators	*n* (Percentage/Range)	National Standard	WHO Standard
Percentage of encounters with one or more antibiotics prescribed	10,402 (23%)	≤15%	20–26%
Percentage of antibiotics prescribed by generic name	369 (55.2%)	100%	100%
Percentage of patients prescribed antibiotics with diagnosis recorded	485 (97%)	85%	100%
Percentage of diagnoses compliant with Uganda Clinical Guidelines (UCG)	367 (73%)		
Percentage of patients receiving antibiotic injections	125 (25%)	≤15%	13.4–24.1
Percentage of antibiotics prescribed consistent with the EMHL	669 (100%)	100%	100%
Percentage of prescriptions in accordance with Uganda Clinical Guideline (UCG)	212 (58%)	100%	100%
Average cost of antibiotics prescribed per patient in USD	$0.5 (0.4–0.8)	$ 0.5–0.9	
Average duration of prescribed antibiotic treatment days	4.8 (3–10)	5–10	5–10

**Table 4 antibiotics-10-00172-t004:** Showing diagnosis for which antibiotic prescriptions were considered noncompliant with the guidelines.

Diagnosis	Frequency (*n* = 155)	Percentage
Malaria	50	32
Helminthiasis	17	11
Gastritis	14	9
Dental conditions	19	12
Viral conditions	14	9
Candidiasis	8	5
Noninfectious conditions	23	15
Others	10	6

Dental conditions included pulpitis, dental caries, and gingivitis; viral conditions were chicken pox, measles, acute diarrhea, common cold, and herpes zoster; noninfectious conditions included dysmenorrhea, lumbago, fatigue, skin allergy/rash, pain, and hernia; others were scabies, wrong antibiotics, or single antibiotic when combination is indicated.

**Table 5 antibiotics-10-00172-t005:** Showing diagnoses that were considered noncompliant with the guidelines.

Diagnosis	Frequency (*n* = 133)	Percentage
Respiratory tract infections	59	44
Upper respiratory tract infections	40	30
Gastroenteritis	15	11
Bacterial infection	10	8
Lower respiratory tract infections	4	3
Others *	5	4

* Dental paraplegia, Sexually Transmitted Infection, breast enlargement.

**Table 6 antibiotics-10-00172-t006:** Showing results for health facility indicators.

Health Facility Indicators	*n* (Percentage/Range)	National Standard	WHO Standard
Existence of standard treatment guidelines (STGs/UCG) for infectious diseases	5 (100%)	100%	100%
Existence of an approved hospital essential medicines list	0 (0%)	100%	100%
Availability of a set of key antibiotics in the facility stores on the day of the study	51 (68%)	100%	100%
Average number of days that a set of key antibiotics is out of stock in the review	8(5–15)	0%	0%
Expenditure on antibiotics as a percentage of total facility medicine costs	$3200 (38%)		
Availability of diagnostic laboratory facility (Complete blood count)	2 (40%)	100%	100%

## Data Availability

The data for the study will be available upon reasonable request.

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
