# Peer review of "An Evaluation of Antibiotic Prescribing Practices in a Rural Refugee Settlement District in Uganda"

_antibiotics, 2021, doi:10.3390/antibiotics10020172_

Round 1

Reviewer 1 Report

Antibiotic resistance (AR) is a major healthcare-related problem that has been rising around the globe. The misuse of antibiotics is one of the main reasons that cause antibiotic resistance. In this present study, the authors assessed antibiotic prescribing practices in a rural district in Uganda.  This cross-sectional investigation study,  which involved a retrospective review of 500 outpatient prescriptions from five Health facilities. Overall, the manuscript looks good and can be considered for possible publication in the antibiotics journal as the topic and the content is highly related to the journal.

Minor comments

It would be better if authors co-relate the antibiotic prescribing pattern and with the diseases, with respiratory, urinary tract infections, and then malaria in the discussion section.

Author Response

Thank you for your valuable comments. Here is our response to your comment.

Comments

Response

Page no.

It would be better if authors co-relate the antibiotic prescribing pattern and with the diseases, with respiratory, urinary tract infections, and then malaria in the discussion section.

We have added these statement.

These correlates with the prescribing patterns of the antibiotics. For example, the UCG recommends amoxicillin for non-severe pneumonia and ceftriaxone for severe pneumonia while ciprofloxacin is recommended for urinary tract infections like acute cystitis and prostatis.

Page 5

Reviewer 2 Report

Bonniface M et al, presented this very interesting paper about antimicrobial prescribing practices in a rural refugee settlement District in Uganda. I think it is very important to provide data about antibiotic prescription in low resource settings in order to build adequate stewardship programs, that could not be automatically imported by other models.

Considering the draft content I would suggest to put the "Materials and Methods" section after the introduction as it helps readers to understand better the results and the discussion. Besides I invite authors to define abbreviation the first time they are used, and then to use only abbreviations.

Moreover I would suggest some minor correction considering spelling and sentence structure, as following:

-Line 80: please spell the abbreviation EMHSL

-Line 84: please put the abbreviation UCG

-Line 97: see comment for line 80

-Line 99: HCIV, the explanation of this abbreviation appear in line 249, even if is mentioned in line 99 for the first time

-Line 147-149: please reformulate these sentence to improve understanding

-Line 158: use PPF abbreviation

-Line 161-162: please reformulate this sentence as for example "and because they were medicines of choice for the prescribers for the majority of health conditions"

-Line 164: please eliminate "towards to these antibiotics"

-Line 182: use only UCG

-Line 186: use only UCG

-Line 194: eliminate "as"

-Line 204: just put EMHSL

-Line 249: see comment line 99

-Line 251: please spell OPD

-Line 259: just put WHO

-Line 281 and 291: please just put EMHSL

Author Response

Thank you for your valuable comments.
